# Sexual risk behaviour, sexual victimisation, substance use and other factors related to depression in men who have sex with men in Wenzhou, China: a cross-sectional study

Xiaohong Pan,[1] Runhua Li,[1,2] Qiaoqin Ma,[1] Hui Wang,[1] Tingting Jiang,[1] Lin He,[1] Shidian Zeng,[3] Dayong Wang,[3] Zhenmiao Ye,[3] Haishen Zhu,[4] Dongshe Zhao,[4] Shichang Xia[1]

XP and RL contributed equally.

[1]Department of HIV/AIDS and STDs Control & Prevention, Center for Disease Control and Prevention of Zhejiang Province, Hangzhou, Zhejiang, China
[2]Department of Epidemiology & Biostatistics, School of Medicine, Ningbo University, Ningbo, Zhejiang, China
[3]Department of HIV/AIDS and STDs Control & Prevention, Wenzhou Center for Disease Control and Prevention, Wenzhou, Zhejiang, China
[4]Department of HIV/AIDS and STDs Control & Prevention, Lucheng Center for Disease Control and Prevention, Wenzhou, Zhejiang, China

**Correspondence to**
Dr Shichang Xia;
shchxia@cdc.zj.cn

## ABSTRACT

**Objectives** Men who have sex with men (MSM) are a marginalised population at high risk for a variety of mental health problems that may mutually facilitate HIV transmission. We assessed factors associated with depressive symptoms among MSM, which can provide some guidance for future mental health interventions with the MSM population for prevention of HIV infection and transmission.

**Methods** A cross-sectional study was conducted in Wenzhou city, China using respondent-driven sampling (RDS) between December 2013 and June 2014. A face-to-face questionnaire survey was employed to collect information about mental and psychosocial health conditions and sexual behaviour characteristics among MSM. Bivariate and multivariable logistic regression analyses were used to identify factors associated with major depressive symptoms.

**Results** Of the 454 participants in the study, prevalence of major depressive symptoms was 34.6%. During the past 6 months, 85% had anal sex with men, and rate of consistent condom use during anal intercourse was 45.1%. Of the respondents, 16.1% reported suicidal ideation, 62.6% had a history of smoking and 41.4% had a history of drinking alcohol, of which 46.3% reported that they were once drunk. Drug use was reported in 3.7% of the sample. Adult sexual violence from same-sex partners was 7.9%, and 5.1% reported childhood sexual abuse. ARDS-weighted multivariable analysis showed that major depressive symptoms among MSM were associated with inconsistent condom use during anal sex with men, multiple oral male sexual partners, suicidal ideation, adult sexual violence from male partners and being once drunk in the past year.

**Conclusion** We found high levels of depressive symptoms, unprotected sex and suicidal ideation among MSM. Comprehensive intervention strategies that combine psychological, behavioural and social aspects are needed to address the mental health issues of MSM, with special attention given to suicidality and high-risk behaviours in mental health and HIV prevention interventions.

## Strengths and limitations of this study

► The present study obtained the prevalence of major depressive symptoms, sexual risk behaviours, sexual victimisation, suicidal ideation, substance use and elucidated some related syndemic factors to depressive symptoms among men who have sex with men (MSM) in Wenzhou with high HIV prevalence.
► Our study can provide some guidance for mental health and HIV prevention interventions for the MSM population.
► It is unavoidable that MSM might under-report some sensitive issues such as unprotected sex, drug abuse and sexual violence covered in the questionnaire in the rather homophobic and hostile environment of China.
► The causality of depressive symptoms and associated risk factors are not inferred due to the cross-sectional design, and the relevant information is subjected to recall bias, which may have an influence on the magnitude of associations with depression.

## INTRODUCTION

Men who have sex with men (MSM) are a marginalised population facing many challenges including stigma, abuse, social and legal discrimination, and lack of social support,[1–3] which may result in high risk for a variety of mental and psychosocial health problems, such as depression, suicidality, and drug and alcohol abuse.[4–7] A large prospective cohort study showed that syndemics of depressive symptoms, heavy alcohol use and drug use, and childhood sexual abuse (CSA) increased HIV risk and HIV-related risk behaviours.[8] Depression and substance abuse among MSM have been linked to condomless anal intercourse (CAI),[9 10] which was the predominant factor

of HIV acquisition and transmission.[11 12] These interactive factors have resulted in high HIV prevalence and incidence among MSM. It is estimated that MSM accounted for 29.4% of the new HIV infections in China in 2011.[13] In Wenzhou city of China, the prevalence of HIV was estimated to be 22.8% (95% CI, 16.9% to 28.5%) in a previous publication.[14]

A survey conducted in four large American cities showed the prevalence of depressive symptoms among MSM was 29.2%, nearly three times that of the general population.[15] In China, the prevalence of depressive symptoms among MSM has varied from 34.8% to 63.9%,[16–20] which was much higher than 5.3% among the general population.[21] Previous studies showed that factors associated with depression among MSM included race,[22] lower educational level,[23] younger age[24] and marriage to a woman.[25 26] Psychological factors such as sexual victimisation,[27 28] substance abuse[29 30] and HIV or sexual stigma[25] were associated with depressive symptoms among MSM. Some studies showed that engaging in high-risk sexual behaviours was associated with depressive symptoms among the MSM population.[31 32] In China, homosexual behaviour has been subject to prejudice because it continues to be seen by many people as a rejection of China's cultural tradition to marry and have children.[33 34] The marriage between two men was also not admitted in Chinese laws. Chinese MSM have suffered much social stigma and discrimination from the whole society due to their sexual orientation.[33 35 36] Prior research suggests that the pressures tied to maintaining their needs of homosexual behaviour and culturally normative social duties are associated with stress and internalised feelings of blame and low self-worth among Chinese MSM.[36 37] Minority stress theory has been proposed to explain the well-documented prevalence of various mental health problems, such as depression and anxiety symptoms, and suicidal thoughts among MSM.[38]

Few studies have examined the associations between depressive symptoms among MSM and sexual risk behaviours, sexual victimisation and substance use, which is very important to guide mental health and HIV interventions among MSM in China. These previous studies were conducted to examine several independent factors associated with depression and did not comprehensively consider integrating sociodemographic characteristics, high-risk sexual behaviours, psychosocial factors and substance use into their analyses.

Some studies conducted in the USA and in European countries examined depression and associated factors among MSM, whereas such research in China has been limited and lacks comprehensiveness. In the present study, our endeavour was to ascertain the prevalence of depression in Wenzhou city, China and related syndemic factors, including sexual risk behaviours, suicidality, sexual victimisation and substance use. Such knowledge would provide guidance for mental health and HIV prevention interventions for the MSM population.

## MATERIALS AND METHODS
### Study population and sampling

This cross-sectional study was conducted in Wenzhou city, China between December 2013 and June 2014. This baseline survey research was conducted for the purposes of controlling HIV/AIDS new infections among the MSM population in Zhejiang province, China. The study procedures and sampling strategies are described in detail elsewhere.[14] Briefly, participants were recruited using a respondent-driven sampling (RDS) method. Participants who were at least 16 years old, had resided in Wenzhou for at least 3 months, self-reported having had anal sex and/or oral sex in the past 12 months and were willing to participate in the survey met inclusion criteria in this study. First, we selected five initial active 'seed' participants meeting inclusion criteria from the local MSM population. After completing the field investigation and relevant testing, 'seed' participants were provided with three recruitment coupons, and through the incentive measures the 'seed' participants then referred a maximum of three respondents from their social network to participate in the survey. These respondents continued to refer a maximum of three peers to the study and continued to recruit MSM participants until equilibrium was reached. The sample composition was considered to have reached equilibrium when their sociodemographic characteristics were stable, and would not change much for further recruitment of participants (the key sociodemographic characteristics such as age, marital status, education level, monthly income and sexual orientation changed <2% in a subsequent wave recruitment of peers compared with the previous one), irrespective of the characteristics of the initial seeds. All participants received remuneration in the amount of ¥100 (approximately US$16) for participation and another ¥100 as an incentive to refer peers to participate in the study. Finally, five seeds recruited 30, 61, 81, 103 and 174 MSM, respectively, with a total sample of 454 MSM by 14 waves of recruitment. All eligible participants were interviewed by trained staff from the Center for Disease Control and Prevention of Lucheng district, Wenzhou, who also collected the questionnaire about sociodemographics, sexual risk behaviours, psychosocial conditions, HIV-related cognitions and HIV testing history of participants.

## MEASURES
### Mental and psychosocial health conditions
#### Depression

Depressive symptoms were assessed using the Center for Epidemiological Studies Depression Scale (CES-D). The CES-D consists of 20 items designed to assess the experience of depression. Respondents indicate the extent to which each item statement describes their experience during the past week. Response options include the following: 'rarely or none of the time (<1 day)', 'some or a little of the time (1–2 days)', 'occasionally or a moderate amount of the time (3–4 days)' and 'most or all of the

time (5–7 days)'. Responses are on a four-point Likert scale and scored 0–3. Level of depression was categorised as follows: major depressive symptoms (CES-D score of 20 or higher), moderate depressive symptoms (CES-D score of 16–19) and no depressive symptoms (CES-D score of 15 or lower). Thus, higher scores indicate higher levels of depressive symptoms. A total scale score of 20 or higher was used to define major depressive symptoms as the main analysis variable in our study.

### Sexual victimisation

Sexual victimisation included CSA and adult sexual violence from same-sex partners. We defined CSA as the experience of being 'forced or frightened by someone into doing something sexually' with a partner >10 years older than the respondent when the respondent was aged 15 years or younger.[39] Those respondents who reported that they were first forced or coerced to have sex against their will by men when 16 years of age or older were classified as having experienced adult sexual violence. These respondents did not report experiencing CSA (ie, when under the age of 16 years).[40 41]

### Suicidal ideation

Participants were asked: "during the past year, have you ever had thoughts of taking your own life?" The definition of suicidal ideation referred to previous research.[42]

### Use of substances

Substance use included smoking cigarettes, alcohol consumption and use of drugs. Cigarette smoking refers to smoking one or more cigarettes every day for more than a year or smoking more than 300 cigarettes in 3 months or less. Alcohol consumption refers to ethanol intake of 100 g per week such as 250 g of 40% volume of wine consumption or 1 kg of 10% volume of red wine intake. According to the definition, participants were asked whether they had a history of smoking and drinking alcohol, whether they had been drunk in the past year and whether they wanted or needed to reduce drinking in the past year. Drug use was defined as using at least two kinds of the following drugs at least monthly during the past 12 months: sedatives, morphine, mescaline, lysergic acid diethylamide (LSD), opiates, cocaine, marijuana, heroin, ecstasy, other street or club drugs.[43]

### Sexual risk behaviours

Sexual risk behaviours included six items: (1) number of anal intercourse with male partners in the past 6 months, (2) consistent condom use during anal sex with men, (3) number of oral sex with male partners in the past 6 months, (4) consistent condom use during oral sex with men, (5) number of female sexual partners and (6) consistent condom use during vaginal intercourse with women.

### Translation, validation and reliability

Measures were independently translated from English to Chinese by two staff members fluent in both languages.

The surveys were then back-translated into English to ensure that they retained their original meaning and good validation of the translation. The Cronbach's alpha coefficient of the CES-D in a large population from 39 cities in China was 0.90.[44] The Cronbach's alpha coefficient of the CES-D was 0.88 for this sample. The scale is fit for Chinese population with good validation and reliability.[44 45]

### Statistical analysis

Data were entered into EpiData V.3.1 (http://www. epidata.dk/) via double entry. After data cleaning and verification, the data were processed using PASW Statistics 18.0 (SPSS) for statistical analysis. For descriptive analyses, the mean, SD, median and IQR were computed for continuous variables and frequencies for categorical variables. Multicollinearity was diagnosed by examining the variance inflation factors. Variables significantly associated with major depressive symptoms at the level of $p<0.05$ in the univariate logistic analyses were included in the multivariable logistic regression model, controlling for demographic variables (age, marital status, education level, monthly income and sexual orientation) and calculating adjusted ORs (AORs) and 95% CIs. Then, we used the Respondent-Driven Sampling Analysis Tool (RDSAT) V.7.1 (http://www.respondentdrivensampling.org/) to produce individualised weights for adjusting data of different participants with different network sizes.[46–48] The individualised weights were imported into PASW Statistics 18.0 to conduct weighted multivariable logistic regression analyses. Statistical significance was defined as $p<0.05$.

### Ethics statement

The study was approved by the ethical review board of Zhejiang Provincial Center for Disease Control and Prevention. No risk was involved in participating in this study and we protected confidentiality of the participants. A written informed consent was signed by all participants during the survey.

## RESULTS
### Sample characteristics

A total of 454 MSM participated in the study, and the characteristics of the sample are provided in table 1. Of the 454 participants, 55.7% (253/454) were never married and 51.4% (233/453) resided in Wenzhou city for more than 3 years. The age of the respondents ranged from 17 to 76 years (median=33; IQR 26–41). Regarding education level, 86.6% (393/454) had an education of senior school or lower. Roughly 60% (57.3%) of the participants had an income of not less than ¥3000 monthly, 30.8% (140/454) had health insurance and 40.5% reported their sexual orientation as homosexual or gay (table 1).

**Table 1** Sociodemographic characteristics and their associations with depressive symptoms among MSM in Wenzhou, China (n=454)

| Variables | Total, n(%) | Major depressive symptoms, n(%) | OR (95% CI) | p Value |
|---|---|---|---|---|
| Age (years) (median=33; IQR 26–41) | | | | |
| ≤25 | 101 (22.2) | 39 (38.6) | 1.70 (1.01 to 2.84) | 0.044 |
| 26–35 | 168 (37.0) | 68 (40.5) | 1.84 (1.17 to 2.87) | 0.008 |
| 36 and older | 185 (40.7) | 50 (27.0) | 1.00 | |
| Marital status | | | | |
| Never married | 253 (55.7) | 97 (38.3) | 1.00 | |
| Married/divorced/widowed | 201 (44.3) | 60 (29.9) | 0.68 (0.46 to 0.99) | 0.049 |
| Education level | | | | |
| Junior school or lower | 272 (59.9) | 87 (32.0) | 1.00 | |
| Senior school | 121 (26.7) | 50 (41.3) | 1.50 (0.96 to 2.33) | 0.074 |
| College or higher | 61 (13.4) | 20 (32.8) | 1.04 (0.57 to 1.88) | 0.904 |
| Monthly income (¥) | | | | |
| <3000 | 194 (42.7) | 62 (32.0) | 1.00 | |
| 3000 and above | 260 (57.3) | 95 (36.5) | 1.23 (0.83 to 1.82) | 0.310 |
| Duration of residence in Wenzhou (years) | | | | |
| ≤3 | 220 (48.6) | 79 (35.9) | 1.00 | |
| >3 | 233 (51.4) | 77 (33.0) | 0.88 (0.60 to 1.30) | 0.522 |
| Insurance | | | | |
| Insured | 140 (30.8) | 54 (38.6) | 1.00 | |
| Uninsured | 314 (69.2) | 103 (32.8) | 0.78 (0.51 to 1.18) | 0.233 |
| Sexual orientation | | | | |
| Homosexual | 184 (40.5) | 71 (38.6) | 1.34 (0.91 to 1.99) | 0.139 |
| Heterosexual or bisexual or indeterminate | 270 (59.5) | 86 (31.9) | 1.00 | |
| Places to seek for sexual partners | | | | |
| Venues | 261 (59.9) | 85 (32.6) | 1.00 | |
| Internet | 175 (40.1) | 66 (37.7) | 1.25 (0.84 to 1.87) | 0.269 |
| No of MSM which a participant knew in his circle of MSM friends locally (median=7; IQR 3–15) | | | | |
| >7 | 220 (48.5) | 74 (33.6) | 0.92 (0.63 to 1.36) | 0.681 |
| ≤7 | 234 (51.5) | 83 (35.5) | 1.00 | |

MSM, men who have sex with men.

### Depressive symptoms, sexual risk behaviours, suicidal ideation, substance use and sexual victimisation

The median CES-D scale score was 16 (IQR 12–22; range 0–44). The prevalence of major depressive symptoms (CES-D score ≥20) was 34.6% (157/454), whereas 53.5% (243/454) of the participants had moderate to major depressive symptoms (CES-D score ≥16) and 46.5% (211/454) had no depressive symptoms (CES-D score ≤15). Of the 454 participants, 85.0% (386/454) had anal sex with men during the past 6 months and the rate of consistent condom use during anal intercourse was 45.1% (174/386), and 76.7% (348/454) had oral sex with men during the past 6 months and the percentage of consistent condom use during oral sex was 19.8% (69/348). Further, 32.4% reported having anal sex with more than two same-sex partners during the past 6 months, and 31.9% engaged in oral sex during the past 6 months. Suicidal ideation was reported by 16.1% (73/454) of all respondents. History of smoking was reported in 62.6% (284/454) of the participants, and 41.4% (188/454) had a history of drinking alcohol of which 46.3% (87/188) indicated they were once drunk. Only 3.7% (17/454) had used drugs. Regarding sexual victimisation, 7.9% (36/454) had suffered adult sexual violence from same-sex partners and 5.1% (23/454) reported sexual abuse in childhood (tables 2 and 3).

**Table 2** Sexual behaviour and their associations with depressive symptoms among MSM in Wenzhou, China (n=454)

| Variables | Total, n(%) | Major depressive symptoms, n (%) | OR (95% CI) | p Value |
|---|---|---|---|---|
| Age of sexual debut with a man (years) (median=24; IQR 20–30) | | | | |
| ≤22 | 197 (43.4) | 70 (35.5) | 1.08 (0.73 to 1.59) | 0.709 |
| >22 | 257 (56.6) | 87 (33.9) | 1.00 | |
| Anal intercourse with men in the past 6 months | | | | |
| Yes | 386 (85.0) | 139 (36.0) | 1.56 (0.88 to 2.79) | 0.129 |
| No | 68 (15.0) | 18 (26.5) | 1.00 | |
| No of anal intercourse with male partners in the past 6 months (median=2; IQR 1–3) | | | | |
| ≤2 | 307 (67.6) | 98 (31.9) | 1.00 | |
| >2 | 147 (32.4) | 59 (40.1) | 1.43 (0.95 to 2.15) | 0.086 |
| Consistent condom use during anal sex in the past 6 months | | | | |
| Yes | 174 (45.1) | 46 (26.4) | 1.00 | |
| No | 212 (54.9) | 93 (43.9) | 2.18 (1.41 to 3.35) | <0.001 |
| Oral sex with men in the past 6 months | | | | |
| Yes | 348 (76.7) | 124 (35.6) | 1.23 (0.77 to 1.95) | 0.394 |
| No | 106 (23.3) | 33 (31.1) | 1.00 | |
| No of oral male sexual partners in the past 6 months | | | | |
| ≤2 | 309 (68.1) | 97 (31.4) | 1.00 | |
| >2 | 145 (31.9) | 60 (41.4) | 1.54 (1.03 to 2.32) | 0.038 |
| Consistent condom use during oral sex in the past 6 months | | | | |
| Yes | 69 (19.8) | 24 (34.8) | 1.00 | |
| No | 279 (80.2) | 100 (35.8) | 1.05 (0.60 to 1.82) | 0.869 |
| Sex with women in the past | | | | |
| Yes | 348 (76.7) | 115 (33.0) | 0.75 (0.48 to 1.18) | 0.213 |
| No | 106 (23.3) | 42 (39.6) | 1.00 | |
| Age of sexual debut with a woman (years) | | | | |
| ≤22 | 209 (60.1) | 71 (34.0) | 1.11 (0.70 to 1.76) | 0.653 |
| >22 | 139 (39.9) | 44 (31.7) | 1.00 | |
| Sex with women in the past 6 months | | | | |
| Yes | 167 (36.8) | 59 (35.3) | 1.05 (0.71 to 1.57) | 0.798 |
| No | 287 (63.2) | 98 (34.1) | 1.00 | |
| No of female sex partners in the past 6 months | | | | |
| <2 | 407 (90.0) | 137 (33.7) | 1.00 | |
| ≥2 | 45 (10.0) | 19 (42.2) | 1.44 (0.77 to 2.69) | 0.254 |
| Consistent condom use during vaginal sex with women in the past 6 months | | | | |
| Yes | 53 (31.7) | 21 (39.6) | 1.00 | |
| No | 114 (68.3) | 38 (33.3) | 0.76 (0.39 to 1.50) | 0.429 |

MSM, men who have sex with men.

## Factors associated with depressive symptoms

Results of univariate logistic regression analyses are shown in tables 1–4. Young age, never been married, no consistent condom use during anal sex in the past 6 months, number of oral sex male partners in the past 6 months, being once drunk in the past year, suicidal ideation, suffering violence from same-sex partners and experiencing sexual abuse in childhood were associated with depressive symptoms.

Results of the RDS-weighted multivariable analyses are shown in table 4. Major depressive symptoms were associated with inconsistent condom use during anal sex in the past 6 months (AOR=3.62; 95% CI 2.15 to 6.13; p<0.001), more than two oral sexual male partners in the past 6 months (AOR=1.75; 95% CI 1.09 to 2.80; p=0.020), suicidal ideation (AOR=5.09; 95% CI 2.59 to 9.97; p<0.001), suffering adult violence from same-sex partners (AOR=2.57; 95% CI 1.03 to 6.73; p=0.031) and

**Table 3** Substance use, psychosocial factors and their associations with depressive symptoms among MSM in Wenzhou, China (n=454)

| Variables | Total, n(%) | Major depressive symptoms, n(%) | OR (95% CI) | p Value |
|---|---|---|---|---|
| Smoking | | | | |
| Yes | 284 (62.6) | 98 (34.5) | 0.99 (0.67 to 1.48) | 0.966 |
| No | 170 (37.4) | 59 (34.7) | 1.00 | |
| Drinking alcohol | | | | |
| Yes | 188 (41.4) | 64 (34.0) | 0.96 (0.65 to 1.42) | 0.839 |
| No | 266 (58.6) | 93 (35.0) | 1.00 | |
| Being drunk once in the past year | | | | |
| Yes | 87 (46.3) | 39 (44.8) | 2.47 (1.33 to 4.59) | 0.004 |
| No | 101 (53.7) | 25 (24.8) | 1.00 | |
| Wanting or needing to reduce drinking in the past year | | | | |
| Yes | 48 (55.2) | 26 (54.2) | 2.36 (0.99 to 5.67) | 0.054 |
| No | 39 (44.8) | 13 (33.3) | 1.00 | |
| Drug use | | | | |
| Yes | 17 (3.7) | 8 (47.1) | 1.72 (0.65 to 4.55) | 0.275 |
| No | 437 (96.3) | 149 (34.1) | 1.00 | |
| Suicidal ideation | | | | |
| Yes | 73 (16.1) | 50 (68.5) | 5.57 (3.24 to 9.57) | <0.001 |
| No | 381 (83.9) | 107 (28.1) | 1.00 | |
| Adult sexual violence from same-sex partners | | | | |
| Yes | 36 (7.9) | 23 (63.9) | 3.75 (1.84 to 7.63) | <0.001 |
| No | 418 (92.1) | 134 (32.1) | 1.00 | |
| Childhood sexual abuse | | | | |
| Yes | 23 (5.1) | 14 (60.9) | 3.13 (1.32 to 7.41) | 0.009 |
| No | 431 (94.9) | 143 (33.2) | 1.00 | |

MSM, men who have sex with men.

being once drunk in the past year (AOR=2.89; 95% CI 1.36 to 6.14; p=0.006).

## DISCUSSION

This is the first study to explore a comprehensive range of factors for their association with depressive symptoms among MSM in China. Specifically, we investigated risky sexual behaviours, substance use, psychosocial factors including suicidal thoughts and sexual victimisation for their relationship to depressive symptoms. We found high rates of depressive symptoms, unprotected sex and suicidal ideation in our sample. In addition, we found that suffering adult sexual violence from same-sex partners was associated with depressive symptoms.

Our study showed a high rate of depressive symptoms in MSM, higher than the reported prevalence rate of depressive symptoms among the general population in China of 5.3%.[21] A similar increased risk of depressive symptoms has been demonstrated among MSM in the USA, Netherlands, Kenya and South Africa.[15 25 49–51] Due to the prevalence of depressive symptoms among the MSM population greatly exceeding that of the general population, we should lay emphasis on psychosocial assessment of and interventions for MSM.

It is noteworthy to have an association between depressive symptoms and CAI (ie, inconsistent condom use during anal intercourse) in our study, in line with the study conducted by Houston et al.[32] It is possible that MSM with mild to moderate depressive symptoms address their low mood by having unprotected sex, which would increase the sense of intimacy and pleasure.[52–54] Studies have shown that CAI among MSM was the predominant factor in HIV acquisition and transmission.[11 12] Of particular concern were the rising incidence and high prevalence of HIV and lower rate of condom use when engaging in anal sex among MSM in China, specially in Wenzhou.[14 55] Therefore, mental health interventions that focus on reducing high-risk sexual behaviours and improving mental health among MSM may be effective strategies for preventing HIV infection and transmission. In addition, future HIV prevention programmes should consider incorporating mental health screening and

**Table 4** RDS-weighted multivariable analyses of factors associated with depressive symptoms among MSM in Wenzhou, China (n=454)

| Variable | AOR (95% CI) | p Value | RDS-weighted AOR (95% CI) | p Value |
|---|---|---|---|---|
| **Age (years)** | | | | |
| ≤25 | 1.26 (0.62 to 2.54) | 0.526 | 1.17 (0.56 to 2.43) | 0.681 |
| 26–35 | 1.74 (1.01 to 3.01) | 0.048 | 2.43 (1.35 to 4.39) | 0.003 |
| 36 and older | 1 | | 1 | |
| **Marital status** | | | | |
| Never married | 1 | | 1 | |
| Married/divorced/widowed | 0.73 (0.43 to 1.24) | 0.246 | 0.64 (0.36 to 1.12) | 0.119 |
| **Education level** | | | | |
| Junior school or lower | 1 | | 1 | |
| Senior school | 1.55 (0.92 to 2.62) | 0.103 | 2.01 (1.16 to 3.49) | 0.014 |
| College or higher | 1.08 (0.53 to 2.20) | 0.838 | 0.92 (0.45 to 1.88) | 0.825 |
| **Monthly income (¥)** | | | | |
| <3000 | 1 | | 1 | |
| 3000 and above | 0.92 (0.58 to 1.46) | 0.714 | 0.65 (0.40 to 1.03) | 0.068 |
| **Sexual orientation** | | | | |
| Homosexual | 1.05 (0.67 to 1.66) | 0.823 | 0.85 (0.52 to 1.37) | 0.499 |
| Heterosexual or bisexual or indeterminate | 1 | | 1 | |
| **Consistent condom use during anal sex with men in the past 6 months** | | | | |
| Yes | 1 | | 1 | |
| No | 2.13 (1.32 to 3.45) | 0.002 | 3.62 (2.15 to 6.13) | <0.001 |
| **No of oral male sexual partners in the past 6 months** | | | | |
| ≤2 | 1 | | 1 | |
| >2 | 1.44 (0.91 to 2.27) | 0.12 | 1.75 (1.09 to 2.80) | 0.02 |
| **Being drunk once in the past year** | | | | |
| Yes | 2.07 (1.00 to 4.26) | 0.049 | 2.89 (1.36 to 6.14) | 0.006 |
| No | 1 | | 1 | |
| **Suicidal ideation** | | | | |
| No | 1 | | 1 | |
| Yes | 4.54 (2.50 to 8.26) | <0.001 | 5.09 (2.59 to 9.97) | <0.001 |
| **Adult sexual violence from male sexual partners** | | | | |
| No | 1 | | 1 | |
| Yes | 2.25 (1.00 to 5.07) | 0.051 | 2.57 (1.03 to 6.73) | 0.031 |
| **Childhood sexual abuse** | | | | |
| Yes | 1.84 (0.63 to 5.36) | 0.263 | 1.89 (0.34 to 10.57) | 0.467 |
| No | 1 | | 1 | |

AOR, adjusted OR; MSM, men who have sex with men; RDS, respondent-driven sampling.

intervention strategies to improve their effectiveness in promoting sexual health.

In our study, we found a high rate of suicidal ideation and a strong association between suicidal ideation and depressive symptoms. Li et al, in his recent study conducted in Shanghai, China, argued that not feeling a sense of belonging and having little social support may explain the high rates of suicidal ideation in MSM,[56] and the lack of a sense of belonging and social support may cause mental health issues like depressive symptoms. A longitudinal study revealed that lesbian, gay, bisexual and transgender (LGBT) youth with a history of suicide attempt were found to have higher levels of depressive and anxious symptoms.[57] Conversely, MSM with higher levels of self-esteem and greater satisfaction with social support were less likely to have major depression and suicidal ideation.[58] Given

the positive association between major depression and suicidal ideation, a sense of belonging and social support may serve as protective factors as well as mediate the relationship between depression and suicidality. Given the high reported rate of suicidal ideation and depression, mental health centres should provide additional training to clinicians and counsellors on how to care for, screen and manage suicidality in the MSM population. In addition, knowledge of the psychological characteristics of MSM and development of consulting skills in clinic staff and Centers for Disease Control and Prevention (CDCs) related to working with MSM is needed to address their mental health needs, prevent suicidal behaviour and prevent HIV/AIDS infection and transmission. More generally, we should call on society to eliminate discrimination of the MSM population and provide more support and care to prevent suicidal ideation and attempts.

The study revealed that sexual victimisation was associated with depressive symptoms among MSM, consistent with a study by Ratner et al.[40] Sexual violence from male partners and sexual abuse in childhood can have a negative effect on the mental health of MSM, which can increase the risk of HIV infection or other sexually transmitted infections. Consequently, health counsellors and professionals should respond sensitively and give more care to victims of sexual victimisation. For example, clinicians should identify the prevalence of sexual victimisation among MSM, ask sensitive questions in the assessment of mental health and deconstruct prejudicial beliefs about same-sex sexual violence.

Drinking alcohol was reported by 40% and smoking by two-thirds of our sample. Both drinking alcohol and smoking history were not associated with depressive symptoms. However, research indicates a strong association between mental health issues and substance use disorders among the general population in studies of other countries.[59 60] Our study only indicated that being once drunk was associated with depressive symptoms among MSM. Further research is needed to verify the association between drinking alcohol, smoking and depressive symptoms among Chinese MSM. Given there were few drug users in our sample and results of different studies are divergent, the finding that there was no association between drug use and present depressive symptoms in our study should be cautiously interpreted.

Previous studies have shown that high levels of sexual stigma can lead to internalised homophobia and poor social support, all of which are risk factors for mental health conditions such as depression, anxiety, substance disorders and suicidal ideation.[61–64] MSM face social stigma in many settings, which can increase risk for HIV by limiting access to care.[65 66] Thus, developing antistigma interventions and providing support that reduce homophobia and biphobia and encourage LGBT population rights, thereby reducing the amount of stigma, marginalisation and discrimination that MSM face in the first place would be very important in the future public health programmes. It is essential to change social attitudes gradually and reduce stigma and discrimination towards MSM populations. Health providers and MSM community organisations are valuable in providing social and emotional support to reduce stigma, alleviate mental disorder and promote healthy behaviours. Besides, family members and friends from outside the MSM community also offer social resources after MSM disclose their sexual identity. Media campaigns against homophobia may encourage broader acceptance of LGBT issues by portraying MSM in a non-discriminatory and normal manner.

There are some limitations to our study. First, these data were collected via self-report, and MSM might have concerns about their privacy and under-report some sensitive information in the rather homophobic and hostile environment of China. As such, this method may have resulted in an underestimation of the true prevalence of sensitive behaviours, such as unprotected sex, drug abuse and sexual violence. Second, the causality of depressive symptoms and associated factors is not inferred due to the cross-sectional design, and the relevant information is subjected to recall bias, which may have an influence on the magnitude of associations with depression. Third, some important psychosocial factors, such as self-esteem, social support and negative social relationships that might mediate or moderate the associations between depressive symptoms and related factors were not included in the research design. In addition, stigma related to HIV and sexual stigma are important factors associated with depression[25] and were not examined. Fourth, different measures of depressive symptoms, such as the Self-Rating Depression Scale and Beck Depression Inventory may have provided results that differ from our study. Fifth, very few respondents with known HIV-positive status might report not knowing about their status before completing the study questionnaires for the incentive, which had an impact on the study result to some extent because depressive symptoms are very high and common among diagnosed HIV-infected people.[67 68] Thus, longitudinal research with a larger sample size, employing more in-depth and complex measures of psychosocial and behavioural variables, and including such psychosocial factors as social support and social relationships, are needed to confirm our findings.

In response to our findings, mental health and substance abuse screening and counselling should be integrated into HIV prevention and treatment programmes for the MSM population in China. Mental health issues are generally not given enough emphasis in the training of HIV/AIDS counsellors in China, and we have had to provide supplementary training to staff from clinics and CDCs. More comprehensive intervention strategies that combine psychological, behavioural and social aspects are needed to address the existing mental health issues, with special attention paid to suicidality and high-risk behaviours among MSM.

**Acknowledgements** We would like to thank Editage (www.editage.cn) for English language editing.

**Contributors** SX, XP and QM were involved in the study design. HW, TJ, LH, SZ, DW, ZY, HZ and DZ performed the experiments. RL, XP and SX analysed the data. RL, XP and SX wrote the paper. XP, SX and QM critically revised the manuscript. All authors read and approved the final manuscript.

**Funding** Key Project on Social Development among S&T Major Project of Zhejiang Province, China (2013C03047-1) and General Program on Medicine and Health of Zhejiang Province, China (2016KYA066).

**Competing interests** None declared.

**Patient consent** Obtained.

**Ethics approval** The study was approved by the ethical review board of Zhejiang Provincial Center for Disease Control and Prevention.

**Provenance and peer review** Not commissioned; externally peer reviewed.

**Data sharing statement** No additional data are available.

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
