## [Reviewer comments · BMJ Open]

ARTICLE DETAILS

TITLE (PROVISIONAL)	Sexual risk behavior, sexual victimization, substance use, and other factors related to depression in men who have sex with men in Wenzhou, China: a cross-sectional study
AUTHORS	Pan, Xiaohong Li, Runhua Ma, Qiaoqin Wang, Hui Jiang, Tingting He, Lin Zeng, Shidian Wang, Dayong Ye, Zhenmiao Zhu, Haishen Zhao, Dongshe Xia, Shichang

VERSION 1 - REVIEW

REVIEWER	Joseph De Santis, PhD
REVIEW RETURNED	24-Aug-2016

GENERAL COMMENTS	This is an excellent, important, and well-written manuscript. There are only a few minor areas that I believe require attention. These include: 1) I did not see anywhere in the manuscript where the authors talked about translation of the instruments, validation of the translation, etc.2) Could the authors report the reliability coefficients of the CES-D with this sample? Could this be compared to the original or any other studies with Chinese populations? It was a pleasure to review your important work!
---

REVIEWER	Gregory Armstrong
REVIEW RETURNED	25-Aug-2016

GENERAL COMMENTS	Thanks for the opportunity to review this manuscript. The authors report on analysis of data from a survey with data collected from MSM in China using a complex survey design called respondent driven sampling. The authors have provided useful descriptive information on the prevalence of depression. However, the multivariate analyses should be improved and the paper re-submitted. The analyses did not account for the complex sampling design. There are many papers now that detail appropriate
---

	methods to be used when analysing respondent driven sampling, including software that has been created for this purpose, but the authors don't seem to have engaged with this. If they don't do this, then they are essentially reporting on data from a convenience sample, and this undermines all the effort of obtaining the data using this method. Additionally, Table 5 presents the final multivariate model but only seems to report the variables that retained statistical significance. There are other variables (e.g. marital status) that were significant in univariate modelling and would have been included in the model, yet aren't reported on in this table, and we have no way of knowing exactly what variables went into this final model. I think the authors could have created a series of regression models with depression as a predictor variable and selected indicators of sexual behaviours, HIV knowledge, etc as outcome variables. These models could have consistently controlled for confounding from obviously important socio-demographic variables, like age, sexual orientation, income, marital status, places to seek partners, etc.
--	--

REVIEWER	M. Reuel Friedman
REVIEW RETURNED	25-Sep-2016

GENERAL COMMENTS	This is an interesting study looking at factors associated with symptoms of major depression among MSM in Wenzhou City, China. Generally the writing is clear and the manuscript as a whole indicates that depression among MSM in this locale is an unfolding epidemic that needs further attention by public health providers. However there are some issues that need to be addressed before publication. 1) While the authors present a sufficient background for data on depression among MSM, the studies they cite are mostly focused on US populations and even there limited background is provided to understand why depression may occur, e.g. as a sequelae of stigma and discrimination. The introduction and discussion sections would be greatly improved with additional text about the social policies regarding MSM in China, and how social status might influence the development of mental health concerns in this population. If it exists, some description about the social situation of MSM in Wenzhou City specifically would also enrich the introduction and discussion sections. 2) On a related but slightly different note, it is not enough to suggest that we develop interventions to help MSM cope with the mental health conditions that arise from experiences of minority stress, discrimination, and marginalization. Some discussion about the public health importance of developing anti-stigma interventions and supporting structural initiatives that reduce homophobia and biphobia and encourage LGBT rights, thereby reducing the amount of stigma, marginalization, and discrimination that MSM face in the first place, would enrich the discussion section. 3) On line 26, p. 4, suggest removing "group sex" as the sample in the cited study was not representative of MSM as a whole. Also, condomless anal intercourse (CAI) should be used instead of unprotected anal intercourse (UAI) throughout.
--

4) On line 36, p. 4, suggest replacing "up to 22.8%" with "estimated to be 22.8%": also please add the 95% CI for this point prevalence estimate.

5) On line 45, p. 4, suggest replacing "among MSM varied" with "among MSM has varied".

6) A greater case for including "HIV-related cognitions" should be made in the introduction. Without greater justification for what HIV-related cognitions are, and why they might be linked to depressive symptoms, I would have to recommend removing these measures from the analysis.

7) Throughout the manuscript, when discussing the results of these specific analyses, "depression" should be replaced by "depressive symptoms". CES-D does not assess depression per se, rather symptoms of it. For example, scores over 15 indicate the presence of depressive symptoms, not of depression. This needs to be clarified throughout.

8) Some questions in the method section about RDS sampling in this manuscript. How did the authors assess how equilibrium was reached (p. 6, line 37)? The statement that "5 seeds recruited 30, 61, 81, 103, and 174 MSM" doesn't make sense since each seed only had three coupons so should be clarified. Were the results controlled for by recruitment source, e.g. weighted to compensate for the sample being recruited in a non-random way? It does not appear to be the case here, so would recommend that the authors more accurately describe their sampling as a snowball process that was not weighted in analysis (e.g., it's essentially a convenience sample).

9) On p. 7, line 34-35: recommend replacing "certain" with "major", and "possible" with "moderate" unless the authors can provide sources for how they qualify depressive symptoms using the language they have.

10) For measures related to substance use: how is an "excessive amount of drug use" defined and can the authors provide a justification for this measure? Why is caffeine on this list? What is wheat carlin? I am having trouble understanding how the substance use measure could be considered appropriate, it needs much more justification than what is provided.

11) Have the items related to sexual victimization and suicidal ideation been previously used? Suggest citing sources if these measures have been used before; if they have not, more justification about why standard measures were not used would help justify these measures.

12) Under sexual risk behaviors as a measure, why was consistent condom use during oral sex measured? Very few MSM engage in oral sex with condoms. What kind of sex (anal, vaginal, or oral) is described in consistent condom use during sex with women?

13) See my comment above: to my mind, including "HIV-related cognitions" do not make intuitive sense to me in this analysis, and the measures for these are not well-described. What is "influence of social perception on sexual behaviors of MSM population"? Why

	should a participant be able to estimate HIV prevalence among MSM? How is regular testing defined? If you have a possibility of HIV infection...what if some of these participants were already HIV-positive? If HIV status was not assessed, this is a major limitation, as countless studies have shown that HIV infection can be a confounder for depression. 14) In statistical analysis, what is "double entry"? 15) In statistical analysis section, some description of the hypotheses (e.g. likely correlates of depression) would be useful. 16) In the results section, each entry related to sex (e.g., anal sex, oral sex, etc) needs to have a qualifier: sex with men? Sex with women? What is the gender of the partners described in these sexual activities? Were transgender/third-gender partners assessed? (If not that should be a limitation.) 17) Results from the multivariable analyses should be always described as AOR, not OR, in the results section. 95% CI should always accompany OR and AOR estimates in the results. Also, throughout: "multivariable" should replace "multivariate." While this manuscript has some flaws, most of them appear to be minor, and I hope that the authors can respond to the comments above in a way that can significantly improve this manuscript, as the results are valuable.
--	---

VERSION 1 – AUTHOR RESPONSE

Reviewer #1: This is an excellent, important, and well-written manuscript. There are only a few minor areas that I believe require attention. These include:

1) I did not see anywhere in the manuscript where the authors talked about translation of the instruments, validation of the translation, etc.

2) Could the authors report the reliability coefficients of the CES-D with this sample? Could this be compared to the original or any other studies with Chinese populations?

Response: Thank the reviewer for the high appraisal to our manuscript.

1) Measures were independently translated from English to Chinese by two staff members fluent in both languages. The surveys were then back-translated into English to ensure that they retained their original meaning and good validation of the translation.

2) We used the Cronbach's alpha coefficient for assessing the reliability. The Cronbach's alpha coefficient was 0.88 for this sample. The Cronbach's alpha coefficient of the CES-D in a large population from 39 cities in China was 0.90. (Zhang J, Wu Z, Fang G, et al. Development of the Chinese age norms of CES-D in urban area[J]. Chinese Mental Health Journal. 2010, 24(2): 139-143). The scale is fit for Chinese population with good validation and reliability. (Zhang J, Wu Z, Fang G, et al. Development of the Chinese age norms of CES-D in urban area[in Chinese]. Chinese Mental Health Journal. 2010, 24(2): 139-143. Zhang Y, Jia C, Fang Z, et al. Study on depression symptom and reliability and validity of CES - D used in rural residents in Shandong province [in Chinese]. Chin J Public Health 2008;24:473-5.).

Reviewer #2:

Thanks for the opportunity to review this manuscript. The authors report on analysis of data from a survey with data collected from MSM in China using a complex survey design called respondent

driven sampling. The authors have provided useful descriptive information on the prevalence of depression.

1) However, the multivariate analyses should be improved and the paper re-submitted. The analyses did not account for the complex sampling design. There are many papers now that detail appropriate methods to be used when analysing respondent driven sampling, including software that has been created for this purpose, but the authors don't seem to have engaged with this. If they don't do this, then they are essentially reporting on data from a convenience sample, and this undermines all the effort of obtaining the data using this method.

Response: Thank the reviewer for the good comment on the analytical method for data. The method of the analysis might be not very rigorous if the data were not weighted by adjusting the social network size of every participant in data. So we use the software RDSAT 7.1 (<http://www.respondentdrivensampling.org/>) to produce individualized weights for conducting the multivariate analyses again.

2) Additionally, Table 5 presents the final multivariate model but only seems to report the variables that retained statistical significance. There are other variables (e.g. marital status) that were significant in univariate modelling and would have been included in the model, yet aren't reported on in this table, and we have no way of knowing exactly what variables went into this final model.

Response: According to your good comment, we included the variable "marital status" and other socio-demographic variables like "age", "educational level", "income", and "sexual orientation" besides variables significantly associated with the outcome variable at the level of $P < .05$ in the univariate logistic analyses into the multivariate model to control for confounding from socio-demographic variables.

(3) I think the authors could have created a series of regression models with depression as a predictor variable and selected indicators of sexual behaviours, HIV knowledge, etc as outcome variables. These models could have consistently controlled for confounding from obviously important socio-demographic variables, like age, sexual orientation, income, marital status, places to seek partners, etc.

Response: Thanks for the reviewer's suggestion. We have created a series of regression models with depression as a predictor variable and selected indicators of condomless anal sex, oral sex, and substance use as outcome variables controlled for confounding from obviously important socio-demographic variables including age, sexual orientation, monthly income, marital status and education level according to your good comment. We found that condomless anal sex and being once drunk in the past year were still associated with depression. The results of the series of regression models are not demonstrated into the tables in our study in order to not destroy the main framework and contents of the paper.

Reviewer: 3

This is an interesting study looking at factors associated with symptoms of major depression among MSM in Wenzhou City, China. Generally the writing is clear and the manuscript as a whole indicates that depression among MSM in this locale is an unfolding epidemic that needs further attention by public health providers. However there are some issues that need to be addressed before publication.

Response: First, thank the reviewer for good comments on our manuscript very much.

1) While the authors present a sufficient background for data on depression among MSM, the studies they cite are mostly focused on US populations and even there limited background is provided to understand why depression may occur, e.g. as a sequelae of stigma and discrimination. The introduction and discussion sections would be greatly improved with additional text about the social policies regarding MSM in China, and how social status might influence the development of mental health concerns in this population. If it exists, some description about the social situation of MSM in Wenzhou City specifically would also enrich the introduction and discussion sections.

Response: First, thank the reviewer for good comments on our manuscript very much. In fact, China does not have special social policies for MSM. But, we have added some description about additional text about the social cultural tradition regarding MSM in China, and how the social cultural tradition

might influence the development of mental health concerns in this population (line 89-99 in the introduction). The social situation of MSM in Wenzhou City which is influenced very deeply by Chinese traditional culture should be similar to other areas.

2) On a related but slightly different note, it is not enough to suggest that we develop interventions to help MSM cope with the mental health conditions that arise from experiences of minority stress, discrimination, and marginalization. Some discussion about the public health importance of developing anti-stigma interventions and supporting structural initiatives that reduce homophobia and biphobia and encourage LGBT rights, thereby reducing the amount of stigma, marginalization, and discrimination that MSM face in the first place, would enrich the discussion section.

Response: We have added some discussion about the public health importance of developing anti-stigma interventions and supporting structural initiatives in the discussion part (line 325-340 in the discussion part).

3) On line 26, p. 4, suggest removing "group sex" as the sample in the cited study was not representative of MSM as a whole. Also, condomless anal intercourse (CAI) should be used instead of unprotected anal intercourse (UAI) throughout.

Response: According to the suggestions, we have removed "group sex". We have replaced "unprotected anal intercourse (UAI)" with "condomless anal intercourse (CAI)" throughout.

4) On line 36, p. 4, suggest replacing "up to 22.8%" with "estimated to be 22.8%": also please add the 95% CI for this point prevalence estimate.

Response: According to the suggestions, we have replaced "up to 22.8%" with "estimated to be 22.8% (95% CI, 16.9 to 28.5%)"

5) On line 45, p. 4, suggest replacing "among MSM varied" with "among MSM has varied".

Response: According to the suggestions, we have replaced "among MSM varied" with "among MSM has varied".

6) A greater case for including "HIV-related cognitions" should be made in the introduction. Without greater justification for what HIV-related cognitions are, and why they might be linked to depressive symptoms, I would have to recommend removing these measures from the analysis.

Response: According to your suggestion, I have removed these measures for what HIV-related cognitions are from the analysis.

7) Throughout the manuscript, when discussing the results of these specific analyses, "depression" should be replaced by "depressive symptoms". CES-D does not assess depression per se, rather symptoms of it. For example, scores over 15 indicate the presence of depressive symptoms, not of depression. This needs to be clarified throughout.

Response: According to the suggestions, we have replaced "depression" with "depressive symptoms" when discussing the results of these specific analyses throughout.

8) Some questions in the method section about RDS sampling in this manuscript. How did the authors assess how equilibrium was reached (p. 6, line 37)? The statement that "5 seeds recruited 30, 61, 81, 103, and 174 MSM" doesn't make sense since each seed only had three coupons so should be clarified. Were the results controlled for by recruitment source, e.g. weighted to compensate for the sample being recruited in a non-random way? It does not appear to be the case here, so would recommend that the authors more accurately describe their sampling as a snowball process that was not weighted in analysis (e.g., it's essentially a convenience sample).

Response: When the key socio-demographic characteristics such as age, marital status, education level, monthly income and sexual orientation changed <2% in a subsequent wave recruitment of peers compared with the previous one, we think that the equilibrium was reached. Details that describe how to judge a stable status are added into the manuscript (line 131-136 in the materials and

methods part). In fact, our study adopted RDS method rather than snowball sampling. We have conducted weighted multivariate analyses by weighting the sample data using RDSAT software to produce individualized weights by adjusting social network size of every responder.

9) On p. 7, line 34-35: recommend replacing "certain" with "major", and "possible" with "moderate" unless the authors can provide sources for how they qualify depressive symptoms using the language they have.

Response: According to the suggestions, we have replaced replacing "certain" with "major", and "possible" with "moderate".

Response:

10) For measures related to substance use: how is an "excessive amount of drug use" defined and can the authors provide a justification for this measure? Why is caffeine on this list? What is wheat carlin? I am having trouble understanding how the substance use measure could be considered appropriate, it needs much more justification than what is provided.

Response: In our study, "excessive amount of drug use" mean "using at least two of these drugs at least monthly during the past 12 months". In fact, the definition of the measure of drug abuse was not very rigorous, it should call "drug use" rather than "drug abuse". (Friedman MR, Stall R, Silvestre AJ, et al. Effects of syndemics on HIV viral load and medication adherence in the multicentre AIDS cohort study AIDS. 2015 Jun 1;29(9):1087-96.) We have replaced "drug abuse" rather than "drug use" in the revised manuscript.

Drug use included the following drugs: sedatives, morphine, mescalinum, LSD, opiates, cocaine, marijuana, heroin, ecstasy, other street or club drugs. "wheat carlin" refers to "mescalinum" and we are sorry it is a error of translation, and "caffeine" refers to "morphine", which is a error of spelling. We are really sorry for having trouble understanding to you again.

11) Have the items related to sexual victimization and suicidal ideation been previously used? Suggest citing sources if these measures have been used before; if they have not, more justification about why standard measures were not used would help justify these measures.

Response: Sexual victimization includes childhood sexual abuse and adult sexual violence in our study. We defined childhood sexual abuse as the experience of being "forced or frightened by someone into doing something sexually" with a partner more than 10 years older than the respondent when the respondent was aged 15 years or younger. (Paul JP, Catania J, Pollack L, Stall R. Understanding childhood sexual abuse as a predictor of sexual risk-taking among men who have sex with men: the Urban Men's Health Study. Child Abuse Negl. 2001;25:557-584.)

Those respondents who reported that they were first forced or coerced to have sex against their will by men when 16 years of age or older were classified as having experienced adult sexual violence. These respondents did not report experiencing childhood sexual abuse (i.e. when under the age of 16 years). (Ratner PA, Johnson JL, Shoveller JA, et al. Non-consensual sex experienced by men who have sex with men: prevalence and association with mental health. Patient Educ Couns. 2003 Jan;49(1):67-74. Paul JP, Catania J, Pollack L, Stall R. Understanding childhood sexual abuse as a predictor of sexual risk-taking among men who have sex with men: the Urban Men's Health Study. Child Abuse Negl. 2001;25: 557-584.)

The question on suicidal ideation asked: "during the last 6 months, have you ever had thoughts of taking your own life?". The definition of suicidal ideation referred to previous research. (O'Carroll PW, Berman AL, Maris RW, et al. Beyond the Tower of Babel: a nomenclature for suicidology. Suicide Life Threat Behav 1996; 26: 237-252.)

12) Under sexual risk behaviors as a measure, why was consistent condom use during oral sex measured? Very few MSM engage in oral sex with condoms. What kind of sex (anal, vaginal, or oral) is described in consistent condom use during sex with women?

Response: Many studies suggest that the oral sex without condoms has the possibility of transmitting HIV, so engaging in oral sex without condoms is also a kind of sexual risk behavior needed to be measured. (Edwards S, Carne C. Oral sex and the transmission of viral STIs. Sex Transm Infect.

1998;74(1):6-10. Rothenberg RB, Scarlett M, del Rio C, Reznik D, O'Daniels C. Oral transmission of HIV. AIDS. 1998;12(16):2095-105.) Actually, in our study, the sex describing consistent condom use during sex with women referred to the vaginal sex and we have replace "sex with women" with "vaginal sex with women"

13) See my comment above: to my mind, including "HIV-related cognitions" do not make intuitive sense to me in this analysis, and the measures for these are not well-described. What is "influence of social perception on sexual behaviors of MSM population"? Why should a participant be able to estimate HIV prevalence among MSM? How is regular testing defined? If you have a possibility of HIV infection...what if some of these participants were already HIV-positive? If HIV status was not assessed, this is a major limitation, as countless studies have shown that HIV infection can be a confounder for depression.

Response: According to your suggestion above, we have removed the measures of "HIV-related cognitions" such as "influence of social perception on sexual behaviors of MSM population", "estimate of HIV prevalence among MSM", "knowing about ART for HIV/AIDS" and "self-reported possibility of HIV infection" because they were not measured by better methods with greater justification, and they were not understandable well in the paper. Regular testing was defined as taking a HIV test every 3 months, 6 months or every year. Regular testing was not significantly associated with depressive symptoms, so we also remove the measure of regular testing and other testing variables. All respondents recruited in our study did not know their HIV status before completing the study questionnaires, so we did not assess the HIV status of MSM in the sample.

14) In statistical analysis, what is "double entry"?

Response: "Double entry" means "the data from the original questionnaires were keyed into the EpiData by two investigators" to check the accuracy of the data entry.

15) In statistical analysis section, some description of the hypotheses (e.g. likely correlates of depression) would be useful.

Response: We are really very sorry that we cannot totally understand your meaning. Because some description of the hypotheses (correlates of depression) have been placed in the introduction part. (line 100-106)

16) In the results section, each entry related to sex (e.g., anal sex, oral sex, etc) needs to have a qualifier: sex with men? Sex with women? What is the gender of the partners described in these sexual activities? Were transgender/third-gender partners assessed? (If not that should be a limitation.)

Response: We have added a qualifier tied to anal sex, oral sex in the corresponding place of the manuscript.

17) Results from the multivariable analyses should be always described as AOR, not OR, in the results section. 95% CI should always accompany OR and AOR estimates in the results. Also, throughout: "multivariable" should replace "multivariate."

Response: According to the suggestions, we have replaced replacing "OR" with "AOR" in the results from the multivariable analyses and "multivariable" are replaced by "multivariate."

VERSION 2 – REVIEW

REVIEWER	Mackey R. Friedman
REVIEW RETURNED	08-Nov-2016

GENERAL COMMENTS	The authors should be commended for their thorough revision of their previous submission.
---

	I have only one further suggestion: that the authors state in their Methods (sample section) that participants were of unknown HIV status. (This could also be discussed in other sections of the manuscript, as it is an important population to study.) Thank you for the opportunity to review this manuscript.
--	--

REVIEWER	Artemis Koukounari
REVIEW RETURNED	06-Apr-2017

GENERAL COMMENTS	Thank you for giving me the opportunity to read this very interesting article by Xiaohong P, Runhua L et al. I have read it with great interest and I am satisfied with how the authors addressed the reviewers comments, particularly the statistical ones. I do have an objection with a reviewer's comment requesting to change all terms from multivariable to multivariate. The following publication by Hidalgo & Goodman also confirms that this request is not quite reasonable: Am J Public Health. 2013 January; 103(1): 39–40. Multivariate or Multivariable Regression?, so I would please request that the authors change all the relevant terms again back to multivariable. In line 290 a space is needed for the words 'and' and 'trangender' Finally, I think in line 295 authors mean protective factor and thus I would request to change this instead of protector factor.
---

VERSION 2 – AUTHOR RESPONSE

Reviewer: 3

I have only one further suggestion: that the authors state in their Methods (sample section) that participants were of unknown HIV status. (This could also be discussed in other sections of the manuscript, as it is an important population to study.)

Response: Thank the reviewer for the good suggestion. We have already added some limitations in the discussion part (line 354-358) about HIV infection can be a confounder for depression.

Reviewer: 4

Please leave your comments for the authors below.

I do have an objection with a reviewer's comment requesting to change all terms from multivariable to multivariate. The following publication by Hidalgo & Goodman also confirms that this request is not quite reasonable: Am J Public Health. 2013 January; 103(1): 39–40. Multivariate or Multivariable Regression?, so I would please request that the authors change all the relevant terms again back to multivariable.

In line 290 a space is needed for the words 'and' and 'trangender'.

Finally, I think in line 295 authors mean protective factor and thus I would request to change this instead of protector factor.

Response:

Thank the reviewer for the important suggestions.

We have changed all the relevant terms "multivariate" again back to "multivariable".

We have added a space between the words 'and' and 'transgender'.

We have replaced "protector factor" with "protective factor" correctly.